# Assessing Predictive Factors of COVID-19 Outcomes: A Retrospective Cohort Study in the Metropolitan Region of São Paulo (Brazil)

**DOI:** 10.3390/medicina57101068

**Published:** 2021-10-06

**Authors:** Juliana Neide Amato, Paula Midori Castelo, Ferla Maria Simas Bastos Cirino, Guilherme Meyer, Luciano José Pereira, Luís Cláudio Sartori, Natália Simões Aderaldo, Fernando Capela e Silva

**Affiliations:** 1Department of Pharmaceutical Sciences, Universidade Federal de São Paulo (UNIFESP), Diadema 09913-030, Brazil; amato.ju@gmail.com (J.N.A.); paula.castelo@unifesp.br (P.M.C.); nattaderaldo@gmail.com (N.S.A.); 2Diadema Municipal Health Department, Diadema 09911-160, Brazil; ferlacirino@gmail.com (F.M.S.B.C.); guilhermemey@gmail.com (G.M.); kausartori@uol.com.br (L.C.S.); 3Department of Health Sciences, Universidade Federal de Lavras (UFLA), Lavras 37200-900, Brazil; lucianojosepereira@ufla.br; 4Department of Medical and Health Sciences, School of Health and Human Development, University of Évora, 7000-671 Évora, Portugal

**Keywords:** SARS-CoV-2, COVID-19, social determinants of health, diabetes mellitus, obesity

## Abstract

*Background and Objectives:* The aim of this retrospective cohort study was to search individual, sociodemographic and environmental predictors of COVID-19 outcomes. *Materials and Methods:* A convenience sample of 1036 COVID-19 confirmed patients (3–99 years, mean 59 years; 482 females) who sought treatment at the emergency units of the public health system of Diadema (Brazil; March–October 2020) was included. Primary data were collected from medical records: sex, age, occupation/education, onset of symptoms, presence of chronic diseases/treatment and outcome (death and non-death). Secondary socioeconomic and environmental data were provided by the Department of Health. *Results:* The mean time spent between COVID-19 symptom onset and admission to the health system was 7.4 days. Principal component analysis summarized secondary sociodemographic data, and a Poisson regression model showed that the time between symptom onset and health system admission was higher for younger people and those from the least advantaged regions (availability of electricity, a sewage network, a water supply and garbage collection). A multiple logistic regression model showed an association of age (OR = 1.08; 1.05–1.1), diabetes (OR = 1.9; 1.1–3.4) and obesity (OR = 2.9; 1.1–7.6) with death outcome, while hypertension and sex showed no significant association. *Conclusion:* The identification of vulnerable groups may help the development of health strategies for the prevention and treatment of COVID-19.

## 1. Introduction

At the end of 2019, the highly transmittable severe acute respiratory syndrome coronavirus 2 (SARS-CoV-2) appeared in Wuhan, China [1]. The virus spread rapidly across the planet, and to date (27 July 2021), the Johns Hopkins Coronavirus Resource Center has reported more than 195,199,879 confirmed cases and 4,175,769 deaths all over the world. More specifically, in Brazil, diagnosed cases have surpassed 19,707,662 while the number of deaths is over 550,502 [2].

Currently, certain clinical conditions are reported to cause greater susceptibility to complications and a severe evolution of COVID-19, such as cancer, chronic kidney disease, COPD (chronic obstructive pulmonary disease), Down syndrome, heart diseases (heart failure, coronary artery disease and cardiomyopathies), immunocompromised state, obesity, pregnancy, sickle cell disease, smoking and type 2 diabetes mellitus [3]. However, socioeconomic status may represent an additional risk, since socially disadvantaged populations are more likely to be exposed, become sick and die compared to their favored counterparts [4], especially in low- and middle-income countries such as Brazil [5].

Social inequalities and inequities are health determinants [6,7]. These factors should be considered when evaluating COVID-related data to increase the understanding and provide support to health prevention and intervention measures [4]. People from lower socioeconomic groups live in crowded houses, perform work activities that do not allow working from home (increasing contact with co-workers), use public transportation and may not have access to adequate personal protective equipment [8]. Additionally, these people are more susceptible to stress situations at work, burnout syndrome and unemployment, which may reduce immune function and disrupt inflammatory responses [4]. In this context, a simplistic analysis of clinical conditions at the individual level is insufficient to explain health problem outcomes, since social contexts create stratification, differential exposure to harmful conditions and differential vulnerability [9].

Thus, the aim of the present study was to investigate individual, socioeconomic and environmental predictive factors of COVID-19 outcomes using primary and secondary data to identify the most vulnerable groups in order to help government institutions develop target health strategies for the prevention and treatment of COVID-19 patients.

## 2. Materials and Methods

### 2.1. Ethics and Study Design

This retrospective cohort study was carried out using primary and secondary data from the city of Diadema (Latitude: 23°41′11″ S, Longitude: 46°37′24″ W), located in the metropolitan region of the city of São Paulo, Brazil. The present research protocol was approved (10 November 2020) by the Human Research Ethics Committee of the Universidade Federal de São Paulo (UNIFESP, Brazil), protocol number CAAE 39531920.2.0000.5505.

### 2.2. Patients and Primary Data Collection

This study included a convenience sample of 1036 individuals who sought care at the emergency units of the municipality and received a COVID-19 positive diagnosis in the public health system of Diadema between March and October 2020. Data were extracted both from the available primary medical records (*n* = 515) and from the Municipal Health Department archives (*n* = 1036).

Diadema is one of 39 municipalities that are part of the Metropolitan Region of São Paulo city, which has a territorial extension of 30.73 km^2^, per capita income of BRL 565 (currency of Brazil, USD 108.71) and 405,596 inhabitants; of those, 12.3% are aged 60 years old or over [10]. The percentage of households covered by a water supply, garbage collection and sanitary sewage network is 99.43, 99.61 and 96.55%, respectively [10].

The São Paulo Social Responsibility Index (IPRS) combines economic variables, mortality and schooling rates to generate indicators of wealth, longevity and schooling, respectively, which are expressed on a scale from 0 to 100, in which 100 represents the best situation and 0 the worst [11]. According to this index, Diadema is considered an unequal municipality, presenting high levels of wealth (score 42), but with low levels of social longevity (score 66) and education (score 56).

Diadema has a network of twenty municipal public health establishments (Basic Public Health Units), four Emergency Care Public Hospitals, one Municipal Public Hospital and one State Public Hospital, according to data provided by the Municipal Department of Health in 2020.

Public and private units in Brazil (primary care units, private clinics, call centers, emergency units, specialized services in safety engineering and occupational medicine) must compulsorily inform all cases of flu syndrome through an online system (e-SUS Notify and SIVEP-Gripe), which is also used in the case of hospitalization and death due to severe acute respiratory syndrome (SARS). Notifications must be made within 24 h of the initial suspicion of the case or death. The cases of participants diagnosed with COVID-19 that occurred at the beginning of the outbreak, that is, before the legislation that defined the compulsory notification of COVID-19 cases, were notified individually by health professionals of the municipality, and information was sent to epidemiological surveillance. Moreover, these institutions must notify the results of molecular or immunological diagnostic tests for SARS-CoV-2 [12,13,14].

The Health Information Sector of the Municipal Health Department of Diadema sought notifications of COVID-19 through internal archives and access to these official information systems (e-SUS-Notify and SIVEP-Gripe), providing an anonymous database of confirmed cases of COVID-19 by laboratorial criteria. These individuals sought treatment at the emergency units of the public health system from March 2020 to October 2020, and the evolution of cases was part of the records. We had access to the following information: date of notification, sex (male or female), age (in years), Basic Health Unit that was defined by the municipality according to the citizen’s residence address, notifying unit (name of the notifying institution), evolution of the disease (recovery, death, missing), final classification (positive or negative for COVID-19), type of diagnostic examination (RT-PCR or rapid COVID-19 tests), date of onset of symptoms (defined as the day when any symptoms were noticed by the patient). The final sample comprised 1036 symptomatic COVID-19 confirmed individuals (482 females), aged between 3 and 99 years old (average age 59).

Additional information was obtained from medical records of each individual using electronic medical records, including the presence of previous comorbidities and use of medication. Out of 1036 patients, medical records were retrieved for 515 individuals who were under medical follow-up before COVID-19 infection. Primary medical data were not extracted from all the participants due to a lack of availability, missing data and inaccurate records, among other problems.

### 2.3. Secondary Socioeconomic Data Extraction

Secondary socioeconomic, demographic and environmental information was extracted from two different reports using the e-SUS APS [15], in accordance with territorial units within the areas covered by each Basic Health Unit of the municipality of Diadema. The variables extracted from the ‘Consolidated Report on Domestic and Territorial Registration’ reflect socioeconomic indicators, availability to public services and facilities and household characteristics such as housing situation (own, financed, leased, ceded, occupation, street situation, other), house location (urban, rural), electricity availability (yes, no), type of water supply (piped network, well/spring, water tank, other), destination of sanitary sewage (general network, septic tank, rudimentary pit, direct to a river, lake or sea, open sewer, others), waste destination (collected, burned/buried, open, other) and family income (no income, ¼, ½, 1, 2, 3, 4 or >4 Brazilian minimum wage; Brazilian minimum wage is approximately USD 250,00).

The ‘Consolidated Individual Registration Report’ was also accessed to provide individual variables according to the area of coverage of each Basic Health Unit. These data included: the number of active citizens, sex (male or female), educational level (illiterate, elementary school, middle school, high school), job situation (employer, employee, self-employed with social security, self-employed without social security, retired, unemployed, civil servant/military and others) [15].

Thus, for each participant, the secondary socioeconomic, environmental and demographic information was related to their territorial unit within the area covered by the Basic Health Unit.

### 2.4. Statistical Analysis

Statistical analysis was performed using SPSS 27.0 software considering an alpha level of 5% by an applied statistics spec (PMC). Exploratory statistics consisted of means, standard deviation and percentages. As mentioned above, primary medical data related to comorbidities, current medical treatment and use of medications were gathered from 515 participants from whom complete information was extracted from the health information system (e-SUS); concerning the other 409 participants, only data related to COVID-19 outcome (death/non-death), COVID-19 symptom onset, date of health care system admission and secondary sociodemographic data were available. Only raw data were used in the statistical analysis, and no missing data treatment or data imputation was carried out.

Principal component analysis (PCA) was used to estimate the number of components emerging from sociodemographic and environmental secondary data. Data from each area covered by the Basic Health Unit were grouped regarding unemployment percentages, income (up to 1 minimum wage), schooling (>8 years), private health plan and availability of electricity, a water supply, a sewage network and garbage collection. First, the correlation matrix of the standardized variables was examined, and the number of components to retain was based on eigenvalues and the total of the explained variance. As the components showed moderate correlations, Oblimin rotation was performed. The overall Kaiser–Meyer–Olkin (KMO) measure and Bartlett’s test of sphericity were examined, which are required for a good principal component analysis.

Further, to estimate the time frame between the onset of COVID-19 symptoms and health care system admission, a Poisson regression model was used (*n* = 1036). The independent variables age and sex and the two component scores generated from the PCA were considered as potential independent variables. The adjustment of the final model was based on deviance and an omnibus test.

Information on several risk factors of COVID-19 was initially collected from each individual; due to incomplete and/or missing information, statistical analysis was performed using the most common risk factors reported in the literature, namely, diabetes, hypertension and obesity, for which accurate information was obtained. Thus, a multiple logistic regression model was adjusted using backward stepwise elimination to estimate the probability of death outcome in COVID-19 confirmed patients using primary data from 515 patients and considering the independent variables age, sex, schooling and the presence of diabetes, hypertension and obesity (*n* = 515).

## 3. Results

The demographic and clinical characteristics (both secondary and primary data) of the sample according to death or non-death outcome are shown in Table 1. All individuals were COVID-19 confirmed cases and sought treatment at the emergency units of the public health system of Diadema from March to October 2020. The age range was 3–99 years, with a mean of 59 years. Considering all 1036 participants, the mean time spent between COVID-19 symptom onset and admission to the health care system was 7.4 days.

PCA with Oblimin rotation was run to identify components emerging from sociodemographic and economic secondary data. The suitability of PCA was assessed prior to analysis. The overall Kaiser–Meyer–Olkin (KMO) measure was 0.56, and Bartlett’s test of sphericity was statistically significant (*p* < 0.0001), indicating that the data were likely factorizable. After Oblimin rotation of the factors, PCA revealed two components that explained 64% of the total variance and met the interpretability criterion, as observed in Figure 1 and Table 2. The two components were named environmental aspects (use of private health services, water supply, sewage network and garbage collection) and social aspects (schooling, income and unemployment).

Further, Poisson regression was run to predict the time frame between the onset of COVID-19 symptoms and the entry into the health care system using data from the 1036 participants. The adjusted model included the predictive variables age, sex, Component 1 (environmental aspects) and Component 2 (social aspects) (Table 3). The model showed that age and environmental aspects (use of private health services, water supply, sewage network and garbage collection) were predictive factors related to the time spent seeking treatment, meaning that younger people and those from the least advantaged regions spent more time seeking treatment than their counterparts. The predictors sex and social aspects (schooling, income and unemployment) were not significant.

A logistic regression model was adjusted to estimate the probability of death outcome in individuals diagnosed with COVID-19 using primary data from 515 patients and considering the independent variables age, sex, schooling and the presence of diabetes, hypertension and obesity. The best adjustment included the predictive variables age, diabetes and obesity. According to the model, an increase of 8% in the odds of death outcome was observed for each additional year. Moreover, an OR = 1.9 for individuals with diabetes (95% CI 1.1–3.4) and an OR = 2.9 for individuals with obesity (95% CI 1.1–7.6) were observed. The presence of hypertension, schooling and sex did not associate with death outcome (Table 4). The model achieved a Nagelkerke R^2^ = 33.4% and a predictive accuracy of 73%.

## 4. Discussion

This retrospective cohort study aimed to evaluate individual, sociodemographic and environmental predictors of COVID-19 outcomes using primary and secondary data. As the number of COVID-19 cases continues to increase, the identification of predictive factors may help the development of health strategies to minimize the spread of the disease, severity and death rate [16]. The results show that the time between symptom onset and admission to the health system was higher for younger people and those from the least advantaged regions; in addition, a greater likelihood of dying from COVID-19 among older individuals and those with obesity and diabetes was observed.

The time between symptom onset and admission to the health system was similar to previous studies. Some patients reported fever, cough and muscle pain/fatigue before looking for medical assistance. However, current international recommendations have indicated that patients should stay isolated until presenting respiratory difficulties [17]. In accordance with a previous report, dyspnea occurs in about 55% of infected individuals, and the median time from illness onset to dyspnea is about 8 days (mean 7.8 ± 1.1) [18,19], corroborating the present findings.

Younger people and those from the least advantaged regions spent more time seeking treatment than their counterparts, whereas sex and secondary social aspects such as schooling, income and job status were not significant. People over 50 years of age are more susceptible to severe cases of COVID-19 [16]. The immune system of children and young adults seems to have a strong innate immune response [20,21], which may justify the longer time spent searching for health care services. However, between December 2020 and March 2021, mortality rates of people under 50 years of age increased exponentially [22], due to massive vaccination of the elderly and the emergence of variants.

Relating to sociodemographic and environmental aspects, Brazil has gone through disorderly urban development, and infectious diseases were a public health concern even before the pandemic. The quality of water and basic sanitation together with access to preventive measures is a key determinant in controlling infectious diseases, especially considering low-income citizens [23]. In this scenario, those without a water supply, sewage network and garbage collection were more prone to delaying seeking medical attention. Indeed, the pandemic had a greater impact on people who were already in a situation of social vulnerability, poor housing conditions and difficulty in accessing health services, evidencing the need to improve socioeconomic policies [24]. Surprisingly, secondary social aspects were not associated with health access timing. It was expected that the low level of schooling associated with low income would impact the time elapsed from the beginning of symptoms and the search for medical care. However, it is important to reflect on the vulnerability of specific groups, since the SARS-CoV-2 virus does not show contagious selectivity [24], and the studied population comprised only urban area citizens supported by the Brazilian public health system (SUS).

Eating habits and, consequently, nutrition are fundamental determinants of health. Additionally, it is known, albeit not absolutely, that the social and economic environment, including factors as diverse as family income and social status, employment, levels of education and culture and health literacy, among others, can determine access to a balanced diet based on safe and nutritionally adequate foods. The least advantaged groups living in urban areas may have a diet based on high-energy foods, especially those that are ultra-processed and rich in sugar, and poor consumption of vegetables and fruit when compared to their counterparts [25]. These conditions can enhance the development of certain diseases such as obesity, diabetes, hypertension, cancer and malnutrition. In this way, social inequalities, especially poverty, may affect health and well-being, as less advantaged citizens are more likely to fall ill during their lives because they are more exposed to risk factors, and, again, this increases the risk of SARS-CoV-2 infection.

In the present study, older age and the presence of diabetes and obesity were significantly associated with a greater chance of dying from COVID-19. These results corroborate a recent systematic review which included 17,860,001 patients across 14 countries [26] which stated that the presence of at least one comorbidity increases the severity of COVID-19 [16], as chronic diseases reduce the innate immunity response together with a low-grade inflammatory environment, predisposing patients to a more severe illness. Moreover, a systematic review comprising 87 studies, 35,486 patients and 5867 deaths indicated that diabetes was the best predictor of mortality rates [27,28]. The presence of typical complications of diabetes mellitus such as cardiovascular disease, heart failure and chronic kidney disease also increases COVID-19 mortality [29]. Furthermore, hyperglycemia, found in both type 1 and type 2 diabetes [30], and glycolysis promote SARS-CoV-2 replication and cytokine production in monocytes, resulting in T cell dysfunction and epithelial cell death [31]. In this sense, considering the predictor ‘age’, the results agree with previous findings [18,19] which point to a higher prevalence of comorbidities with advancing age (such as obesity, type 2 diabetes and other chronic diseases) [32].

SARS-Cov-2 infection also involves lung tissue damage. A greater difficulty to compensate for the lack of oxygen through invasive mechanical ventilation in patients with obesity was also reported [33], due to decreased diaphragmatic excursion, decreased expiratory reservoir volume and decreased lung functional capacity [34]. The results of a retrospective cohort study showed that 68.6% of patients required invasive mechanical ventilation, and that disease severity was associated with increased BMI [35]. In an interesting conceptual scheme, which the authors called the ‘Severe COVID-19 obesity tetrad’, Landecho et al. [36] stated that obesity entails an increased pro-inflammatory, prothrombotic state associated with a hormonal rearrangement in a context of decreased cardio-respiratory fitness and limited respiratory capacity. In addition, knowing that obesity itself results in an inflammatory state in metabolic tissues [37], the release of inflammatory cytokines IL-6, IL-8 and TNF-α from infected lung tissues can exacerbate the pro-inflammatory picture in patients [33,36]. Regarding the hormonal changes associated with obesity, with an increase in leptin and a decrease in adiponectin levels, Rebello et al. [38] proposed that leptin may be the link between obesity and its high prevalence as a comorbidity of SARS-CoV-2 infection. Given the high prevalence of obesity in the adult population of the state of São Paulo (approximately 55% of the population has excess weight), this aspect is of great concern [39].

It is important to emphasize the profile of the studied population, which was composed of individuals who sought care at the emergency units, and who were usually the symptomatic ones and the most severe cases when compared to others who sought treatment at the primary care units (who were not included). The design of the study—a retrospective cohort with a convenience sample—is the main limitation of this study, which prevents the generalization of the results and inferences regarding the prevalence of COVID-19 outcomes, also limiting the control for other potential confounders. However, the results provide valuable data that recognize and reinforce the need to identify vulnerable groups and the development of health strategies for the prevention and treatment of COVID-19. More research is needed to elucidate the mechanisms underlying the adverse and exacerbated effects of COVID-19 and what interventions can reduce the hospitalization of individuals with COVID-19 and its risk factors. The results of this work can also help to improve public health strategies to reduce the burden of COVID-19.

## 5. Conclusions

The time between COVID-19 symptom onset and admission to the health system was higher for younger people and those from the least advantaged regions with a lower availability of electricity, a sewage network, a water supply and garbage collection. In addition, a greater chance of dying from COVID-19 was observed for individuals with a higher age, with diabetes (1.9 times) and with obesity (2.9 times).

## Figures and Tables

**Figure 1 medicina-57-01068-f001:**
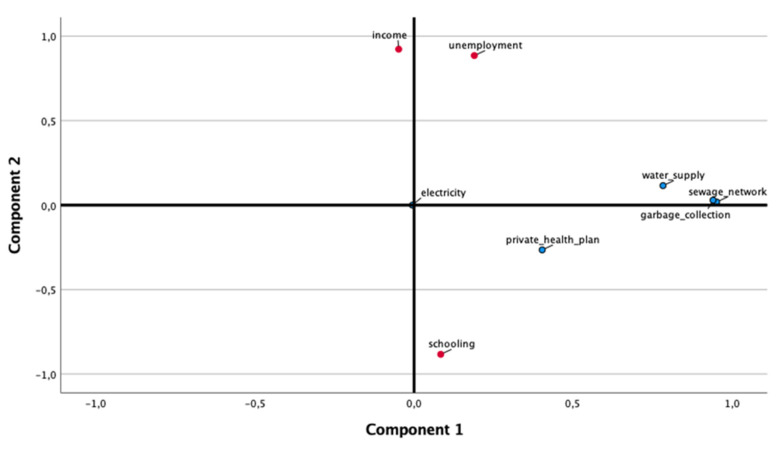
Component plot in rotated space. Component 1: environmental aspects (blue); Component 2: social aspects (red) (*x*-axis: Component 1 scores; *y*-axis: Component 2 scores).

**Table 1 medicina-57-01068-t001:** Demographic and clinical characteristics of a convenience sample of COVID-19 confirmed individuals who sought treatment at the emergency units of the public health system of Diadema (Brazil) from March to October 2020, according to outcome.

Secondary Data (*n* = 1036)		Death Outcome (*n* = 459)	Non-Death Outcome (*n* = 577)
Individual data			
Age (years)	mean (SD)	69.02 (18.5)	50.86
	range	17–99	3–95
Hospitalization (yes)	%	93.46	88.21
Time between COVID-19 symptom onset and health care system admission	mean (SD)	6.80 (6.20)	7.79 (6.18)
Sociodemographic data			
Schooling (>8 years)	%	53.06	55.80
Private health plan	%	21.37	20.99
Electricity	%	99.08	99.03
Water supply	%	99.71	99.73
Sewage network	%	99.27	99.13
Garbage collection	%	99.92	99.91
Income (up to 1 minimum wage)	%	33.82	33.84
Unemployment	%	10.95	10.82
Homeownership	%	58.48	58.31
Urban area	%	99.37	99.38
**Primary Data (*n* = 515)**		**Death Outcome** **(*n* = 251)**	**Non-Death Outcome** **(*n* = 264)**
Individual data			
Age (years)	mean (SD)	69.59 (18.5)	52.44 (17.94)
	Range	17–99	3–95
Schooling (>8 years)	%	55.77	55.26
Hospitalization (yes)	%	93.23	88.26
Time between COVID-19 symptom onset and health care system admission	mean (SD)	6.38 (6.34)	8.16 (6.33)
Diabetes	%	40.64	23.1
Hypertension	%	50.77	36.36
Obesity	%	6.77	6.44

**Table 2 medicina-57-01068-t002:** Component loadings obtained by principal component analysis and Oblimin rotation* of the secondary sociodemographic and economic data.

		Component	
		1	2
		Environmental Aspects	Social Aspects
%	Variance explained	34.9%	28.8%
Schooling (>8 years)			−0.883
Use of private health services		0.403	
Electricity			
Water supply		0.783	
Sewage network		0.952	
Garbage collection		0.941	
Income (up to 1 minimum wage)			0.923
Unemployment			0.885

* Rotation converged in 5 iterations. Only coefficients greater than 0.30 are shown.

**Table 3 medicina-57-01068-t003:** Poisson regression model used to estimate the time frame between the onset of COVID-19 symptoms and the entry into the health care system (*n* = 1036).

Independent Variable	B	Exp (B)	CI (95%) Exp (B)	Wald Chi-Square	*p*-Value
Intercept	2.083	8.030	7.402–8.710	2518.81	0.000
Age	−0.002	0.998	0.997–0.999	7.744	0.005
Sex	0.027	1.027	0.981–1.075	1.302	0.254
Component 1 (environmental aspects)	−0.040	0.961	0.942–0.981	14.103	0.000
Component 2 (social aspects)	−0.002	0.998	0.976–1.022	0.020	0.887

Omnibus test: *p* < 0.001.

**Table 4 medicina-57-01068-t004:** Binomial logistic regression model used to estimate the probability of death of COVID-19 patients using primary data from 515 patients.

Dependent Variable	Independent Variable	B	Exp (B)	CI (95%) Exp(B)	Wald Chi-Square	*p*-Value
Death outcome	Constant	−4.861	-	-	53.006	0.000
	Age	0.076	1.079	1.057–1.101	55.319	0.000
	Obesity	1.060	2.885	1.092–7.620	4.573	0.032
	Diabetes	0.647	1.909	1.088–3.350	5.077	0.024

Omnibus test *p* < 0.001; Hosmer and Lemeshow test *p* = 0.333; Nagelkerke R^2^ = 33.4%; predictive accuracy = 73%.

## Data Availability

Data will be made available upon request.

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
