# Peer review of "Assessing Predictive Factors of COVID-19 Outcomes: A Retrospective Cohort Study in the Metropolitan Region of São Paulo (Brazil)"

_medicina, 2021, doi:10.3390/medicina57101068_

Round 1
Reviewer 1 Report
The paper reviews risk factors for COVID-19. It documents known risk factors including older age, diabetes mellitus and obesity. It seeks to identify additional sociodemographic risk factors in a location with high rates of poverty and social disadvantage. The data tables presented show improved survival in those with more schooling and those with more employment. Those with private health plans were more likely to die. Other factors showed no differences in survival. Covariate models showed that factors associated with poverty were predictive of delayed care for COVID-19. Logistic regression identified only the known factors as predictors of death. It is unclear from this work that delays in care impact outcomes, cost of care or other important factors. It might be that the showing data regarding the need for ICU care, mechanical ventilation or other aspects of care would make a more compelling argument for these social factors having importance in the COVID-19 pandemic.
Author Response
Editor-in-chief, Medicina
Medicina-1352358
Dear Editor,
Thank you for the opportunity to resubmit this revised manuscript.
Please find below the responses to Reviewers. We performed a careful review of the manuscript, hoping that it is now much improved. The changes made are listed and described below, as well highlighted in the text (yellow). In addition, a revision of the grammar/language was performed.
We remain available for further review of our manuscript, if necessary. We hope that the revised version of this manuscript now meets the journal requirements.
With best wishes,
The authors
_______________________________
Reviewer #1
The paper reviews risk factors for COVID-19. It documents known risk factors including older age, diabetes mellitus and obesity. It seeks to identify additional sociodemographic risk factors in a location with high rates of poverty and social disadvantage. The data tables presented show improved survival in those with more schooling and those with more employment. Those with private health plans were more likely to die. Other factors showed no differences in survival. Covariate models showed that factors associated with poverty were predictive of delayed care for COVID-19. Logistic regression identified only the known factors as predictors of death.
It is unclear from this work that delays in care impact outcomes, cost of care or other important factors. It might be that the showing data regarding the need for ICU care, mechanical ventilation or other aspects of care would make a more compelling argument for these social factors having importance in the COVID-19 pandemic.
Answer: Thank you for your suggestion. Indeed, it would be pertinent to include more information concerning individual medical care and other potential confounders to ascertain the impact of each studied predictors on COVID-19 outcomes. Due to the nature and methodology of the study (retrospective cohort), we were not able to evaluate many important aspects, and this is the main limitation of the study. This point was addressed in the last paragraph of the Discussion section (highlighted - yellow), as follows: “The design of the study – retrospective cohort with convenience sample – is the main limitation of this study, which prevents the generalization of the results and the control for other potential confounders on COVID-19 outcomes”.

Reviewer 2 Report
The authors performed a case-control study (with cases being those who died and controls being those who did not die, among patients with Covid-19) to investigate the individual, sociodemographic, and environmental predictors of Covid-19 outcomes using primary and secondary data. They included 1036 patients and evaluated the time between start of Covid-19 symptoms to entry into the healthcare system, evaluated the odds of death in those with higher age, diabetes, obesity, and other risk factors, and studied the correlation of various sociodemographic and environmental factors with Covid-19 outcomes. They found that older age and diabetes were associated with higher probability of dying from Covid-19 and that the time from Covid symptoms to entry into the health system was higher for younger people and those from least advantaged regions with lower availability of electricity, sewage network, water supply, and garbage collection. While this topic is of interest and environmental factors that could be targets of intervention would be good to identify, there are several major concerns with both the design as well as lack of clarity in the presentation of data. Several specific comments are listed below:
-The study design was not initially clear from the abstract; it was evident from the tables (though without legends clearly visible) that the cases were deaths and controls were non-deaths, though this was not explicitly stated or outlined in the methods or abstract
-The authors state that medical records were only available for 515 individuals; they don’t say how missing data was addressed and what proportion of those without records available were either cases or controls (whether there could be misclassification of exposures based on differential missing data)
-The methods used to study clinical risk factors (e.g., obesity and diabetes) were significantly different than those used to study the socioeconomic predictors which is confusing and can be distracting; it seems that this manuscript may best be divided into different papers evaluating different predictors for which there would be complete data available for the entire cohort
-The authors state that Covid-19 cases were available from public health records though Table 1 indicates that 88% of the non-death group was admitted to the hospital (compared with 93% of the death group); these numbers seem markedly high (particularly in the non-death group) which raises concern about the way in which the cases and controls were ascertained
-In-depth discussion of ACE2 seems to detract from the purpose of identifying environmental factors associated with Covid-19
-They state in Table 4 that they are estimating the probability of death but it seems that estimating a specific probability would not be possible from the case-control study design that they reportedly used
-Figure 1 does not indicate what the axis units are
-It does not seem that confounding is adequately addressed; the authors don’t discuss whether they suspect that the specific environmental factors (e.g., sewage network) are associated with Covid-19 outcomes or whether these are indicators of other socioeconomic variables that may be more directly linked to outcomes
Author Response
Editor-in-chief, Medicina
Medicina-1352358
Dear Editor,
Thank you for the opportunity to resubmit this revised manuscript.
Please find below the responses to Reviewers. We performed a careful review of the manuscript, hoping that it is now much improved. The changes made are listed and described below, as well highlighted in the text (yellow). In addition, a revision of the grammar/language was performed.
We remain available for further review of our manuscript, if necessary. We hope that the revised version of this manuscript now meets the journal requirements.
With best wishes,
The authors.
_______________________________
Reviewer #2
-The study design was not initially clear from the abstract; it was evident from the tables (though without legends clearly visible) that the cases were deaths and controls were non-deaths, though this was not explicitly stated or outlined in the methods or abstract.
Answer: Dear Reviewer, thank you for your suggestion. As required, more information was added in the Abstract, as well as throughout the paper concerning the design of the study and the outcomes to improve the reader comprehension (highlighted – yellow).
-The authors state that medical records were only available for 515 individuals; they don’t say how missing data was addressed and what proportion of those without records available were either cases or controls (whether there could be misclassification of exposures based on differential missing data).
Answer: Thank you for our comment. This subject is now clearly addressed in the Methods/Statistics section, as follows: “ As mentioned above, primary medical data related to comorbidities, current medical treatment, and use of medications were gathered from 515 participants from whom complete information was extracted from the health information system (e-SUS); concerning the other 409 participants, only data related to COVID-19 outcome (death/non-death), COVID-19 symptoms onset, date of health care system admission, and secondary socio-demographic data were available. Only raw data were used in the statistical analysis, and no missing data treatment or data imputation has been carried out”.
-The methods used to study clinical risk factors (e.g., obesity and diabetes) were significantly different than those used to study the socioeconomic predictors which is confusing and can be distracting; it seems that this manuscript may best be divided into different papers evaluating different predictors for which there would be complete data available for the entire cohort
Answer: We agree with your opinion; as the objective was to identify biological together with socioeconomic predictors of COVID-19 outcomes, we improved the presentation of the Methods section, and the methodology of primary and secondary data extraction is now detailed in two subsections: “2.2. Patients and primary data collection” and “2.3. Secondary socioeconomic data extraction”.
-The authors state that Covid-19 cases were available from public health records though Table 1 indicates that 88% of the non-death group was admitted to the hospital (compared with 93% of the death group); these numbers seem markedly high (particularly in the non-death group) which raises concern about the way in which the cases and controls were ascertained.
Answer: Indeed, this rates are high but it should be emphasized that this is a convenience sample of participants, and we do not intend to present or discuss the prevalence of the disease or the frequency of Covid-19 outcomes. Thus, we emphasized the design of the study (retrospective cohort) and sampling characteristics (convenience) throughout the paper to ensure a better presentation of the Methods and an accurate understanding of the results and limitations of the study. These details are now included in the Abstract, Table 1, Methods, and Discussion sections (highlighted – yellow).
-In-depth discussion of ACE2 seems to detract from the purpose of identifying environmental factors associated with Covid-19
Answer: We do agree with your suggestion and have removed some explanations that seemed out of the context of the study, focusing the Discussion in the main objective of the study.
-They state in Table 4 that they are estimating the probability of death but it seems that estimating a specific probability would not be possible from the case-control study design that they reportedly used.
Answer: We agree with your concern and have corrected the design of the study, which would be better described as “retrospective cohort” study. Indeed, a convenience sample of 1036 participants and related retrospective data was included and divided into two outcomes: death and non-death (which would also be characterized as ‘nested case-control study’). In this way, the chance of death and non-death outcomes was evaluated considering the available retrospective data gathered from this cohort sample of patients, allowing the interpretation of the likelihood of dying of COVID-19 considering the comorbidities reported in their medical files.
-Figure 1 does not indicate what the axis units are
Answer: As requested, we included the information on X and Y axes of the plot in the Figure 1 caption.
-It does not seem that confounding is adequately addressed; the authors don’t discuss whether they suspect that the specific environmental factors (e.g., sewage network) are associated with Covid-19 outcomes or whether these are indicators of other socioeconomic variables that may be more directly linked to outcomes.
Answer: Indeed, it would be interesting and pertinent to control for these confounding factors; however, due to the nature and methodology of the study (retrospective cohort), we were not able to evaluate many important aspects, and this is the main limitation of the study. This point was addressed in the last paragraph of the Discussion section (highlighted - yellow), as follows: “The design of the study – retrospective cohort with convenience sample – is the main limitation of this study, which prevents the generalization of the results and the control for other potential confounders on COVID-19 outcomes”.

Round 2
Reviewer 2 Report
I appreciate the authors’ revisions and do think that their revisions add some clarity to the study design, methods, and results.
It is still not clear to me what the study design was; while the authors initially stated that it was a case-control study they have now revised this to indicate a retrospective cohort study (in which some people died and some did not die, but people were not specifically identified for the study based on their death vs non-death outcome), which does seem more accurate.
Similarly with the study population, if it is a cohort study as opposed to a case-control study and almost half of the people died (and ~90% were hospitalized), the methods state that this convenience sample included people who sought treatment in the public health system. With these markedly high hospitalization and death rates (even if not the primary purpose of the study), more detail should be provided in the methods regarding how these people were selected, as surely this was not a general sample of all people who sought care. Were these people who were identified in an emergency room or urgent care setting or did it also include people who presented to primary care clinics?
Would still recommend further removing text discussing ACE2 and the pathophysiology of obesity; while interesting and very important, I think that this in some ways detracts from the main intended messages.
The limitations section could be expanded to discuss other limitations and how the generalizability is limited (who are the main people included and how does this differ from the general population with COVID-19?). What are other limitations in missing primary data for some of the cohort, and in sicker patients (with high hospitalization and death rates) being included? Do these people tend to have different sociodemographic and economic characteristics than others and how may this bias results?
Author Response
Editor-in-chief, Medicina
Medicina-1352358
Dear Editor,
Thank you for the opportunity to resubmit this revised manuscript (R2). Please find below the responses to Reviewers. The changes made are listed and described below, as well highlighted in the text (yellow). We remain available for further review of our manuscript, if necessary. We hope that the revised version of this manuscript now meets the journal requirements.
With best wishes,
The authors
_______________________________
Reviewer
- I appreciate the authors’ revisions and do think that their revisions add some clarity to the study design, methods, and results. It is still not clear to me what the study design was; while the authors initially stated that it was a case-control study they have now revised this to indicate a retrospective cohort study (in which some people died and some did not die, but people were not specifically identified for the study based on their death vs non-death outcome), which does seem more accurate.
Answer: We do believe the study design can be classified as retrospective cohort considering the explanation found in the literature: “A retrospective cohort study (also known as a historic study or longitudinal study) is a study where the participants already have a known disease or outcome. The study looks back into the past to try to determine why the participants have the disease or outcome and when they may have been exposed”. Although a case-control study would also fit the design of the present study (“A case-control study is designed to help determine if an exposure is associated with an outcome [i.e., disease or condition of interest]. In theory, the case-control study can be described simply. First, identify the cases [a group known to have the outcome] and the controls [a group known to be free of the outcome]. Then, look back in time to learn which subjects in each group had the exposure[s], comparing the frequency of the exposure in the case group to the control group” [Lewallen and Courtright, 1998]), some authors erroneously consider the case-control design as cross-sectional study. Thus, based on the searched literature, the study may be better designated as “retrospective cohort study”.
- Similarly with the study population, if it is a cohort study as opposed to a case-control study and almost half of the people died (and ~90% were hospitalized), the methods state that this convenience sample included people who sought treatment in the public health system. With these markedly high hospitalization and death rates (even if not the primary purpose of the study), more detail should be provided in the methods regarding how these people were selected, as surely this was not a general sample of all people who sought care. Were these people who were identified in an emergency room or urgent care setting or did it also include people who presented to primary care clinics?
Answer: Thank you for your comment. We agree that further description and explanation are needed. Indeed, the population studied was composed of individuals (usually symptomatic ones) who sought care at the emergency unit. Other individuals could have sought treatment in primary care (in which the number of COVID-19 tests performed is smaller) and in the private sector, but they were not included. More details were added in the text (Abstract, Methods, Results and Discussion/limitations; highlighted – yellow) to better describe the study population.
- Would still recommend further removing text discussing ACE2 and the pathophysiology of obesity; while interesting and very important, I think that this in some ways detracts from the main intended messages.
Answer: As suggested, the Discussion section was revised to emphasize the role of social determinants on Covid-19.
- The limitations section could be expanded to discuss other limitations and how the generalizability is limited (who are the main people included and how does this differ from the general population with COVID-19?). What are other limitations in missing primary data for some of the cohort, and in sicker patients (with high hospitalization and death rates) being included? Do these people tend to have different sociodemographic and economic characteristics than others and how may this bias results?
Answer: As suggested, the limitation section was expanded to better discuss the possible bias in the generalization of the results.
